# SELMA: Learning and Merging Skill-Specific Text-to-Image Experts with Auto-Generated Data

**Jialu Li**[*]    **Jaemin Cho**[*]    **Yi-Lin Sung**    **Jaehong Yoon**    **Mohit Bansal**
UNC Chapel Hill
{jialuli, jmincho, ylsung, jhyoon, mbansal}@cs.unc.edu

https://selma-t2i.github.io

## Abstract

Recent text-to-image (T2I) generation models have demonstrated impressive capabilities in creating images from text descriptions. However, these T2I generation models often fail to generate images that precisely match the details of the text inputs, such as incorrect spatial relationship or missing objects. In this paper, we introduce **SELMA**: **S**kill-Specific **E**xpert **L**earning and **M**erging with **A**uto-Generated Data, a novel paradigm to improve the faithfulness of T2I models by fine-tuning models on automatically generated, multi-skill image-text datasets, with skill-specific expert learning and merging. First, SELMA leverages an LLM's in-context learning capability to generate multiple datasets of text prompts that can teach different skills, and then generates the images with a T2I model based on the prompts. Next, SELMA adapts the T2I model to the new skills by learning multiple single-skill LoRA (low-rank adaptation) experts followed by expert merging. Our independent expert fine-tuning specializes multiple models for different skills, and expert merging helps build a joint multi-skill T2I model that can generate faithful images given diverse text prompts, while mitigating the knowledge conflict from different datasets. We empirically demonstrate that SELMA significantly improves the semantic alignment and text faithfulness of state-of-the-art T2I diffusion models on multiple benchmarks (+2.1% on TIFA and +6.9% on DSG), human preference metrics (PickScore, ImageReward, and HPS), as well as human evaluation. Moreover, fine-tuning with image-text pairs auto-collected via SELMA shows comparable performance to fine-tuning with ground truth data. Lastly, we show that fine-tuning with images from a weaker T2I model can help improve the generation quality of a stronger T2I model, suggesting promising weak-to-strong generalization in T2I models.

## 1 Introduction

Text-to-Image (T2I) generation models have shown impressive development in recent years [61; 59; 50; 30; 57; 85; 10]. Although these approaches can generate high-quality images based on textual inputs, they still struggle to capture all semantics in the given textual prompts, such as failing to compose multiple subjects [85; 22; 40] and generate correct spatial relationships [48].

Many recent works have been proposed to tackle these challenges in text-to-image generation, aiming to enhance the faithfulness of T2I models to textual inputs. One line of research focuses on supervised fine-tuning on high-quality image-text datasets with human annotations [18] or image-text pairs with re-captioned text prompts [65; 5], as shown in Fig. 1 (a). Another line of research is based on aligning T2I models with human preference annotations [82; 52; 21; 33; 76], as shown in Fig. 1 (b). Other

---

[*]equal contribution

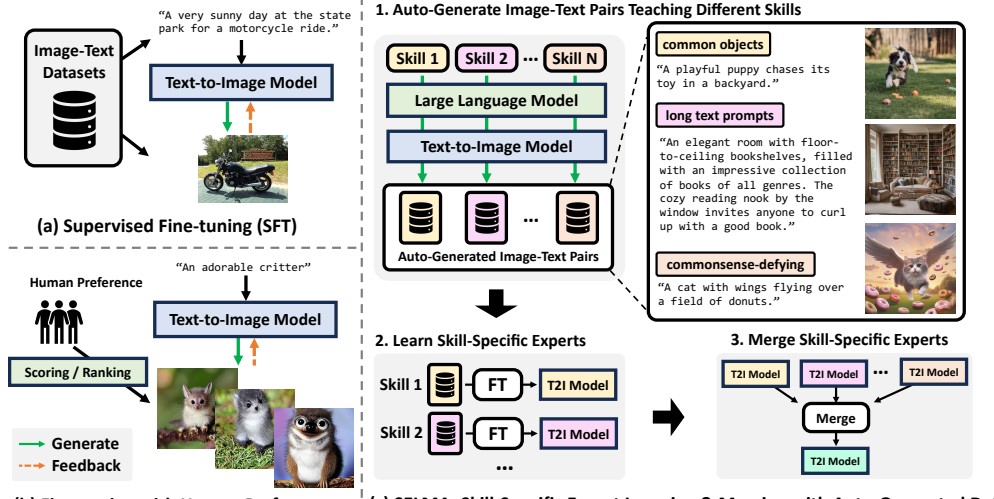

Figure 1: Comparison of different fine-tuning paradigms for text-to-image (T2I) generation models. **(a) Supervised Fine-tuning (SFT)**: a T2I model is trained with image-text pairs from existing datasets. **(b) Fine-tuning with Human Preference (*e.g.*, RL/DPO)**: humans annotate their preferences on images by ranking/scoring in terms of text alignments, and a T2I model is trained to maximize the human preference scores. **(c) SELMA**: instead of collecting image-text pairs or human preference annotations, we automatically collect image-text pairs for desired skills with LLM and T2I model, and create a multi-skill T2I model by learning and merging skill-specific expert models.

works focus on introducing additional layouts or object grounding boxes to guide the generation process [37; 81; 84; 22; 16; 89]. Despite achieving significant improvements in aligning generated images with input textual prompts, the success of these approaches relies on the quality of the layouts created from the textual prompts, the collection of high-quality annotations with human efforts, or the existence of large-scale ground truth data, which involves expensive human annotation.

Motivated by LLMs' impressive text generation capability (given open-ended task instructions and in-context examples), and recent T2I models' capability in generating highly realistic photos (based on text prompts), we investigate an interesting question to further improve the faithfulness of state-of-the-art T2I models: "*Can we automatically generate multi-skill image-text datasets with LLMs and T2I models, to effectively and efficiently teach different image generation skills to T2I models?*" In this paper, we propose **SELMA**: **S**kill-Specific **E**xpert **L**earning and **M**erging with **A**uto-Generated Data, a novel paradigm for eliciting the pre-trained knowledge in T2I models for improved faithfulness based on skill-specific learning and merging of experts. SELMA consists of four stages: (1) collecting skill-specific prompts with in-context learning of LLMs, (2) self-generating image-text samples for diverse skills without the need of human annotation nor feedback from reward models, (3) fine-tuning the expert T2I models on these datasets separately, and (4) obtaining the final model by merging experts of each dataset for efficient adaptation to different skills and mitigation of knowledge conflict in joint training. We illustrate the SELMA pipeline in Fig. 1 (c).

In the first and second stages, we use the LLM and the T2I model to generate skill-specific image-text data. The skills include understanding common objects (*e.g.*, puppy in a backyard), handling long prompts (*e.g.*, an elegant room with floor-to-ceiling bookshelves, filled with an impressive collection of books of all genres. The cozy reading nook by the window invites anyone to curl up with a good book."), and displaying commonsense-defying scenes (*e.g.*, cat flying over sky). We aim to teach diverse generation skills to the same T2I model (*i.e.*, self-learning), so that they can handle different types of prompts. To generate image-text pairs for different skills, we first query GPT-3.5 [46] for prompt generation by using only three skill-specific prompts as in-context examples, and filter the generated prompts with ROUGE-L score to maximize prompt diversity to collect 1K prompts in total (Sec. 3.1). Then we use Stable Diffusion models [59; 50] themselves to generate corresponding images from the prompts (Sec. 3.2). We find that our skill-specific training can help mitigate knowledge conflict when jointly learning multiple skills (see Table 2).

In the third and fourth stages, we fine-tune a T2I model with the collected image-text pairs to teach different skills. However, updating the entire model weights can be inefficient; knowledge conflicts within mixed datasets may also lead to suboptimal performance [42]. Thus, in the third stage, we fine-tune T2I models on these self-generated image-text pairs with parameter-efficient LoRA (low-rank adaptation) modules [27] to create skill-specific expert T2I models (Sec. 3.3). In the fourth stage, to build a joint multi-skill T2I model that can have faithful generations across different skills, we merge the skill-specific experts based on LoRA merging [66; 91] (Sec. 3.4).

We validate the usefulness of SELMA with public state-of-the-art T2I models – a family of Stable Diffusion – v1.4 [59], v2 [59], and XL [50] on two text faithfulness evaluation benchmarks (DSG [14] and TIFA [28]), three human preference metrics (Pick-a-Pic [31], ImageReward [82], and HPS [79]), and human evaluation. Empirical results demonstrate that SELMA significantly improves T2I models' faithfulness to input text prompts and achieves higher human preference metrics. Our final LoRA-Merging model achieves 6.9% improvements on DSG, 2.1% improvements on TIFA, and improves the human preference metrics by 0.4 on Pick-a-Pic, 0.39 on ImageReward, and 3.7 on HPS. Furthermore, we empirically show that the T2I models learned from the self-generated images achieve a performance similar to that of learning from ground-truth images (see Fig. 3). Lastly, we further show that fine-tuning with images from a weaker T2I model (*i.e.*, SD v2) can help improve the faithfulness of a stronger T2I model (*i.e.*, SDXL), suggesting promising weak-to-strong generalization in text-to-image models (see Table 3).

## 2 Related Work

**Training Vision-Language Models with Synthetic Images.** As recent denoising diffusion models [69; 25] have achieved photorealistic image synthesis capabilities, many works have studied using their synthetic images for training different models. Azizi *et al*. [4], Sariyildiz *et al*. [63], Lei *et al*. [34], inter alia, study training image classification models with synthetic images. For image captioning, Caffagni *et al*. [9] use diffusion models to generate images on the captioning data. For training CLIP [53] models, several works use diffusion models to generate images from existing captions [74] or text generated with language models [23]. There is a recent research direction using synthetic images to train image generation models themselves, and we discuss more details in the following paragraph.

**Training Text-to-Image Generation Models with Synthetic Images.** A line of recent works train text-to-image (T2I) generation models with synthetic images generated by the same or other models annotated with human preference scores using reinforcement learning [33; 82; 79; 20; 17; 21] or direct preference optimization (DPO) [55; 76]. While these works show promising results in improving model behavior with human preferences, they require expensive human preference annotations. SPIN-Diffusion [87] proposes using self-play [62; 73], which was successfully adopted in Alphago Zero [67] and language models [13; 88], where the model itself becomes a judge and iteratively compares itself with previous iterations. However, self-play algorithm still relies on a set of ground truth image-text pairs as positive examples for supervision. Concurrent/independent to our work, DreamSync [70] trains a T2I model by first creating text prompts with LLMs, sampling multiple images by the T2I model itself, filtering out images with off-the-shelf scorers, and fine-tuning the model on the resulting synthetic image-text pairs [70]. Unlike DreamSync that depends on image filtering (generating 8 images and taking at most one of them for each text prompt, SELMA generates 1 image for each prompt), significantly improving data generation efficiency by using only 2% of image-text pairs compared with DreamSync. Furthermore, we focus on learning multiple skills with T2I models by learning and merging skill-specific LoRA experts to mitigate knowledge interference across different skills, and we show this approach attains much stronger performance without adding any additional inference cost (see Table 2).

## 3 SELMA: Learning and Merging Text-to-Image Skill-Specific Experts with Auto-Generated Data

We introduce SELMA, a novel framework to teach different skills to a T2I generation model based on auto-generated data and model merging. As illustrated in Fig. 2, SELMA consists of four stages: (1) skill-specific prompt generation with LLM (Sec. 3.1), (2) image generation with T2I Model (Sec. 3.2), (3) skill-specific expert learning (Sec. 3.3), and (4) merging expert models (Sec. 3.4).

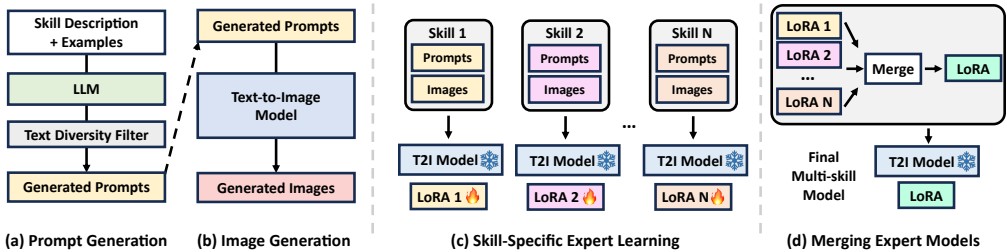

Figure 2: Illustration of the four-stage pipeline of SELMA (Sec. 3).

## 3.1 Automatic Skill-Specific Prompt Generation with LLM

As shown in Fig. 2 (a), we automatically collect skill-specific prompts (that will be paired with images in Sec. 3.2) to fine-tune T2I models in two steps: (1) using large language models (LLMs) to generate prompts with brief skill descriptions and a few example prompts and (2) filtering the generated prompts to ensure their diversity. In the following, we explain the two steps in detail.

**Prompt Generation.** We leverage the in-context learning ability of LLMs to generate additional text prompts that follow similar writing styles (*e.g.*, paragraph style) or acquire models' knowledge in the same domain (*e.g.*, count capability). We manually collect three seed prompts with similar writing styles or acquire similar skills (*e.g.*, spatial reasoning) to the target text prompts. Next, we use these seed prompts as in-context learning examples to query GPT-3.5 (`GPT3.5-turbo-instruct`) [46]. We provide additional instructions that encourage diversity in generated prompts, including object occurrences, sentence patterns, and required skills for the T2I model to generate accurate prompts. The detailed prompt template can be found in the Appendix. During prompt generation, we keep expanding the seed prompts with the generated prompts, and always randomly sample three prompts as in-context learning examples from the seed prompts.

**Prompt Filtering.** To improve the diversity of the collected text prompts, we filter out prompts that are similar to already generated ones. As Taori *et al*. [72] demonstrate that instruction diversity is crucial for improving the instruction following capability of large language models, we follow the same intuition to create diverse text prompts. To ensure the diversity of generated prompts, we first receive a newly generated text prompt from the previous step. Then, we calculate its highest ROUGE-L [38] score with all the previously generated and filtered prompts. Following Taori *et al*. [72], we discard text prompts with ROUGE-L>0.8 to maximize the diversity of generated prompts.

## 3.2 Automatic Image Generation with Text-to-Image Models

As illustrated in Fig. 2 (b), we generate corresponding images for each generated text prompt using the T2I model. We find that existing diffusion-based T2I models are highly effective in learning from their self-generated images, and even benefit from learning from images generated with weaker T2I models (Table 3). It is important to leverage the knowledge that already exists inside the T2I models (learned from web data during pre-training), and hence we aim to extract this knowledge for creating the skill-specific image-text pairs, and use them to improve T2I models' faithfulness (Sec. 3.3).

## 3.3 Fine-tuning with Multiple Skill-Specific LoRA Experts

We efficiently adapt the T2I model to different skills by learning skill-specific Low-Rank Adaptation (LoRA) [27] experts. In LoRA fine-tuning, the updates to the original weights $W_0 \in R^{d \times d}$ is decomposed with two low-rank matrices: $W_0 + \Delta W = W_0 + BA$, where $B \in R^{d \times r}$, $A \in R^{r \times d}$ and $r << d$. For each new dataset, we fine-tune the T2I model with LoRA independently, and this introduces $\mathcal{N}$ skill-specific LoRA experts (as shown in Fig. 2 (c)). In Sec. 5.2, we observe that learning and merging skill-specific experts is more effective than learning a single LoRA across all datasets, by helping the T2I model mitigate knowledge conflicts between different skills [42; 11]. However, using multiple skill-specific experts requires the model to know which expert to use for a given input, and this usually requires user annotations on the skill category of inputs. In the next section, we propose to merge skill-specific experts to efficiently construct a single multi-skill model.

### 3.4 Merging LoRA Expert Models to Obtain a Multi-Skill Model

Recent work of model merging [29; 68; 2; 71; 83] proposes to merge multiple task-specific weights into one, while retaining the original task-specific performances. Moreover, model merging can help mitigate the knowledge conflicts between datasets because we only need to adjust the merging ratios without re-training the task-specific models [86; 56]. Due to these benefits, we extend model merging to learn a final T2I model that can handle multiple skills without knowledge conflicts. Concretely, given $\mathcal{N}$ LoRA experts learned from Sec. 3.3, we merge all LoRA experts into one ($A^{\text{merged}} = \frac{1}{\mathcal{N}} \sum_{n \in \mathcal{N}} A^n$ and $B^{\text{merged}} = \frac{1}{\mathcal{N}} \sum_{n \in \mathcal{N}} B^n$); the resulting single expert can handle all $\mathcal{N}$ skills simultaneously (as shown in Fig. 2 (d)). With this approach, we can reach superior performance over standard multi-task LoRA training and even MoE-LoRA (learning a router with LoRA experts), as shown in Tables 2 and 5, and also eliminate the need to know the skill categories beforehand. Note that while ZipLoRA [66] has demonstrated the use of LoRA merging (merging 2 LoRA modules) in diffusion models, to the best of our knowledge, we are the first to show the effectiveness of LoRA merging on multiple diverse skills (from 5 datasets) in diffusion models.

## 4 Experimental Setup

### 4.1 Evaluation Benchmarks

We evaluate models on two evaluation benchmarks that measure the alignment between text prompts and generated images: **DSG** [14] and **TIFA** [28].

**DSG** consists of 1060 prompts from 10 different sources (160 prompts from TIFA [28], and 100 prompts from each of Localized Narratives [51], DiffusionDB [77], CountBench [47], Whoops [6], DrawText [41], Midjourney [75], Stanford Paragraph [32], VRD [43], PoseScript [19]). Among the ten DSG prompt sources, we mainly experiment with text prompts from five prompt sources that have (1) ground-truth image-text pairs (to compare the usefulness of auto-generated data with ground-truth data) and (2) measuring different skills required in T2I generation (*e.g.*, following long captions, composing infrequent objects). Specifically, we use **COCO** [39] for short prompts with common objects in daily life, **Localized Narratives** [51] for paragraph-style long captions, **DiffusionDB** [77] for human-written prompts that specify many attribute details, **CountBench** [47] for evaluating object counting, and **Whoops** [6] for commonsense-defying text prompts.

**TIFA** consists of 4,081 prompts from four sources, including COCO [39] for short prompts with common objects, PartiPrompts [85] / DrawBench [61] for challenging image generation skills, and PaintSkills [15] for compositional visual reasoning skills.

### 4.2 Evaluation Metrics

We quantitatively evaluate the performance of T2I generation models in text faithfulness and human preference metrics. Specifically, to evaluate text faithfulness, we use VQA accuracy from TIFA [28] and DSG [14]. To evaluate human preference score, we use the PickScore [31], ImageReward [82], and HPS [79] See also Sec. 5.6 for human evaluation. Details can be found in Appendix.

### 4.3 Implementation Details

In the **prompt generation** stage (Sec. 3.1), we use `gpt-3.5-turbo-instruct` [46] to generate text prompts. We collect 1K prompts for each of the five datasets (COCO [39], Localized Narratives [51], DiffusionDB [77], CountBench [47], and Whoops [6]). We refer to the resulting auto-generated datasets as Localized Narrative$^{\text{SELMA}}$, CountBench$^{\text{SELMA}}$, DiffusionDB$^{\text{SELMA}}$, Whoops$^{\text{SELMA}}$, and COCO$^{\text{SELMA}}$, and the resulting combination of 5K auto-generated dataset as DSG$^{\text{SELMA-5K}}$.

In the **image generation** stage (Sec. 3.2), we use the default denoising steps 50 for all models, and the Classifier-Free Guidance (CFG) [26] of 7.5. In the **LoRA fine-tuning** stage (Sec. 3.3), we use 128 as the LoRA rank. **During inference**, we uniformly merge the specialized LoRA experts into one multi-skill expert (Sec. 3.4). More details can be found in Appendix.

Table 1: Comparison of SELMA and different text-to-image alignment methods on text faithfulness and human preference (see Sec. 5.1 for discussion). SELMA achieves the best performance in all five metrics when adapted on different base models (*i.e.*, SD v1.4, SD v2, and SDXL). Best scores for each model are in **bold**.

| Base Model | Methods | Text Faithfulness | | Human Preference on DSG prompts | | |
|---|---|---|---|---|---|---|
| | | DSG$^{mPLUG}$ ↑ | TIFA$^{BLIP2}$ ↑ | PickScore ↑ | ImageReward ↑ | HPS ↑ |
| SD v1.4 [59] | Base model | 67.3 | 76.6 | 20.3 | -0.22 | 23.0 |
| | *(Training-free)* | | | | | |
| | SynGen [58] | 66.2 | 76.8 | 20.4 | -0.24 | 24.5 |
| | StructureDiffusion [22] | 67.1 | 76.5 | 20.3 | -0.14 | 23.5 |
| | *(RL)* | | | | | |
| | DPOK [21] | - | 76.4 | - | -0.26 | - |
| | DDPO [7] | - | 76.7 | - | -0.08 | - |
| | *(Automatic data generation)* | | | | | |
| | DreamSync [70] | - | 77.6 | - | -0.05 | - |
| | **SELMA (Ours)** | **71.3** | **79.5** | **20.5** | **0.36** | **25.5** |
| SD v2 [59] | Base model | 70.3 | 79.2 | 20.8 | 0.17 | 24.0 |
| | **SELMA (Ours)** | **77.7** | **83.2** | **21.3** | **0.72** | **27.5** |
| SDXL [50] | Base model | 73.3 | 83.5 | 21.6 | 0.70 | 26.2 |
| | DreamSync [70] | - | 85.2 | - | 0.84 | - |
| | **SELMA (Ours)** | **80.2** | **85.6** | **22.0** | **1.09** | **29.9** |

## 5 Results and Analysis

### 5.1 Comparison with Different Alignment Methods for Text-to-Image Generation

We compare SELMA with different alignment methods for T2I generation, including training-free methods (SynGen [58], StructureDiffusion [22]), RL-based methods (DPOK [21], DDPO [7]), and DreamSync [70], a concurrent method based on automatic data generation. We experiment with three diffusion-based T2I models (*i.e.*, SD v1.4, SD v2, and SDXL).

**SELMA outperforms other alignment methods for T2I generation.** As shown in Table 1, SELMA consistently improves faithfulness and human preference metrics for all three backbones. Specifically, on SD v1.4, SELMA improves the baseline by **2.9%** in TIFA, **4.0%** in DSG, **0.2** in PickScore, **0.58** in ImageReward, and **2.5** in HPS score. Furthermore, SELMA achieves significantly higher performance than other baselines, including the RL-based methods (DPOK/DDPO), which require annotated human preference data, and DreamSync, a concurrent/independent work based on a larger auto-generated dataset (*i.e.*, 28K text prompts; SELMA uses 5K text training prompts in total), and image filtering (*i.e.*, generating 8 images and taking at most one of them for each text prompt; SELMA only generates 1 image for each prompt). Besides, on SD v2 and SDXL, SELMA shows larger improvement in text faithfulness (*i.e.*, **7.4%** improvement on DSG for SD v2, and **6.9%** on DSG for SDXL), demonstrating the effectiveness of SELMA.

### 5.2 Effectiveness of Learning & Merging Skill-Specific Experts

We compare (1) separately learning multiple LoRA experts on different auto-generated datasets followed by merging and (2) training a single LoRA on a mixture of datasets. For this, we experiment with our five auto-generated image-text pairs: Localized Narrative$^{SELMA}$, CountBench$^{SELMA}$, DiffusionDB$^{SELMA}$, Whoops$^{SELMA}$, and COCO$^{SELMA}$ (see Sec. 4.3 for details).

**Learning & merging skill-specific LoRA experts is more effective than single LoRA on multiple datasets.** Table 2 shows that the LoRA models trained separately on each of the five automatically generated datasets (*No.1.* to *No.5.*) can improve the overall metric over the baseline SD v2 – 70.3%, while the degree of improvements is different for each metric (*e.g.*, 76.4% for fine-tuning with Localized Narrative$^{SELMA}$, and 73.0% for fine-tuning with DiffusionDB$^{SELMA}$). However, training multiple skills simultaneously with a single LoRA (*No.6.* to *No.7.*) tends to degrade performance as more datasets are incorporated. This indicates that the T2I model struggles with LoRA to accommodate distinct skills and writing styles from different datasets. A similar phenomenon has been reported in LLaVA-MoLE [11], where the knowledge conflict between multiple datasets can degrade the performance. We're the first to show this knowledge conflict across different skills also

Table 2: Comparison of single LoRA and LoRA Merging (see Sec. 5.2 for discussion). We use SD v2 as our base model and train models with our automatically generated image-text pairs. DATA$^{SELMA}$: auto-generated image-text pairs where prompts are generated with LLMs with three prompt examples from DATA that are not included in DSG test prompts (see Sec. 4.3 for details). *LN: Localized Narratives; CB: CountBench; DDB: DiffusionDB.* Best/2nd best scores are **bolded**/underlined.

| No. | Model | Auto-Generated Training Dataset | | | | | Text Faithfulness | | Human Preference on DSG | | |
|---|---|---|---|---|---|---|---|---|---|---|---|
| | | LN$^{SELMA}$ *(Paragraph)* | CB$^{SELMA}$ *(Count)* | DDB$^{SELMA}$ *(Real Users)* | Whoops$^{SELMA}$ *(Counter-Factual)* | COCO$^{SELMA}$ *(Common Objects)* | DSG$^{mPLUG}$ | TIFA$^{BLIP2}$ | PickScore | ImageReward | HPS |
| 0. | SDv2 | | | | | | 70.3 | 79.2 | 20.8 | 0.17 | 24.0 |
| 1. | | ✓ | | | | | 76.4 | 81.4 | 20.9 | 0.56 | 26.2 |
| 2. | | | ✓ | | | | 76.0 | 81.4 | 20.8 | 0.46 | 25.7 |
| 3. | | | | ✓ | | | 73.0 | 81.2 | 20.9 | 0.46 | 25.8 |
| 4. | + Single LoRA | | | | ✓ | | 73.0 | 80.7 | 20.8 | 0.44 | 25.3 |
| 5. | | | | | | ✓ | 76.0 | 81.3 | 20.9 | 0.47 | 25.6 |
| 6. | | ✓ | ✓ | ✓ | | | 75.1 | 81.5 | 20.7 | 0.37 | 24.8 |
| 7. | | ✓ | ✓ | ✓ | ✓ | ✓ | 74.4 | 80.2 | 20.6 | 0.35 | 24.9 |
| 8. | + LoRA Merging | ✓ | ✓ | ✓ | | | 76.9 | 82.9 | 21.2 | 0.65 | 27.3 |
| 9. | | ✓ | ✓ | ✓ | ✓ | ✓ | **77.7** | **83.2** | **21.3** | **0.72** | **27.5** |

exists in diffusion models. We find that merging multiple skill-specific LoRA experts (*No.8.* and *No.9.*) achieves the best performance in both text faithfulness and human preference, demonstrating that merging LoRA experts can help mitigate the knowledge conflict between multiple skills.

## 5.3 Effectiveness of Auto-Generated Data

In this section, we investigate the effectiveness of our automatically generated data by comparing them with ground truth data. We fine-tune SD v2 model using ground truth data from Localized Narratives, CountBench, DiffusionDB, Whoops, and COCO, sampling 1K image-text pairs from each dataset and fine-tuning specialized LoRA experts accordingly.

**Fine-tuning with auto-generated data can achieve comparable performance to fine-tuning with ground truth data.** As shown in Fig. 3, we observe that fine-tuning with either auto-generated or ground truth data improves from baseline SD v2 performance – 70.3%, when evaluated on the DSG benchmark. Surprisingly, fine-tuning with the generated data via SELMA outperforms the use of ground truth data in most cases, leading to a DSG accuracy improvement of **4.0%** with Localized Narrative style prompts, **1.0%** with Count-Bench style prompts, **1.9%** with Dif-

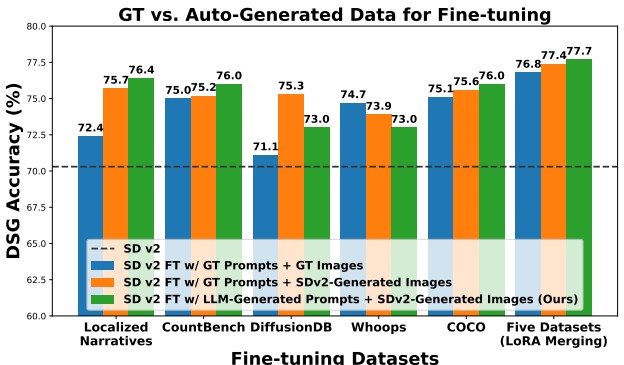

Figure 3: DSG accuracy of SD v2 fine-tuned with different image-text pairs.

fusionDB style prompts, and **0.9%** with COCO style prompts. In short, our approach results in an average improvement of 1.2% brought by fine-tuning only auto-generated data without any need for human-collected ground truth text-image pairs, suggesting that diffusion-based text-to-image models may benefit from the diversity of self-generated images. Furthermore, we investigate whether the improvement is brought by text prompt or image quality. We generate images with SD v2 based on 1K ground truth captions, and fine-tune specialized LoRA experts accordingly. We observe that in most cases, using generated images works better than ground truth images (*e.g.*, Localized Narrative), suggesting T2I models can generate images with comparable alignment as ground truth images. Besides, learning from our LLM-generated captions achieves comparable performance with learning from ground truth captions, suggesting the effectiveness of our text prompt collection process. Lastly, we also show in Appendix that fine-tuning with LLaMA3 [1] generated prompts also improve T2I models' faithfulness in generation, demonstrating that our proposed SELMA is compatible to different LLM-based prompt generator.

Table 3: Comparison of different image generators for creating training images. In addition to using the same model being trained as an image generator, we also experiment with using a smaller model as an image generator (No. 4.). SDXL is bigger/stronger than SD v2. See Sec. 5.4 for discussion.

| No. | Base Model | Training Image Generator | Text Faithfulness | | Human Preference on DSG | | |
|-----|-----------|--------------------------|-------------------|---|-------------------------|---|---|
| | | | DSG$^{mPLUG}$ ↑ | TIFA$^{BLIP2}$ ↑ | PickScore ↑ | ImageReward ↑ | HPS ↑ |
| 1. | SD v2 | - | 70.3 | 79.2 | 20.8 | 0.17 | 24.0 |
| 2. | SDXL | - | 73.3 | 83.5 | 21.6 | 0.70 | 26.2 |
| 3. | SD v2 | SD v2 | 77.7 | 83.2 | 21.3 | 0.72 | 27.5 |
| 4. | SDXL | SD v2 | **81.3** | 83.8 | 21.5 | 0.78 | 28.8 |
| 5. | SDXL | SDXL | 80.2 | **85.6** | **22.0** | **1.09** | **29.9** |

## 5.4 Weak-to-Strong Generalization

In previous experiments, we demonstrate the interesting self-improving capabilities of T2I models, where the training images were generated by the same T2I model. Here, we delve into the following research question: *"Can a T2I model benefit from learning with images generated by a weaker model?"*. The problem of *weak-to-strong* generalization was initially explored in the context of LLMs [8; 60], referred to as superalignment, which involved training GPT-4 [45] using responses generated by a weaker agent, such as GPT-2.

**Weaker T2I models can help stronger T2I models.** As shown in Table 3, fine-tuning SDXL with generated images from SD v2 (*No.4.*) remarkably enhances performance over the SDXL baseline (*No.2.*) in both text faithfulness and human preference. In addition, this approach achieves competitive performance compared with fine-tuning SDXL with SDXL-generated images (*No.5.*), indicating a promising potential for weak-to-strong generalization in diffusion-based T2I generation models. To the best of our knowledge, this is the first work to find promising improvements in the weak-to-strong generalization for text-to-image diffusion models.

## 5.5 Comparison with Prompt Generation with LLaMA3

In this section, we demonstrate that our proposed paradigm is compatible with different prompt generator LLMs. Specifically, we experiment with LLaMA3 (8B) [1], a publicly available open-source LLM and compare the results with GPT-3.5 (gpt-3.5-turbo-instruct) based setups described in the main paper. With both LLMs, we generate five sets

Table 4: DSG and TIFA accuracy of SDXL fine-tuned with prompt data generated with LLaMA3 and GPT-3.5.

| Model | Prompt Generator | Image Generator | DSG$^{mPLUG}$ ↑ |
|-------|------------------|-----------------|-----------------|
| SDXL | - | - | 73.3 |
| SDXL | LLaMA3 | SDv2 | 78.0 |
| SDXL | LLaMA3 | SDXL | 78.6 |
| SDXL | GPT3.5 | SDv2 | 81.3 |
| SDXL | GPT3.5 | SDXL | 80.2 |

of skill-specific prompts, and each set contains one thousand skill-specific prompts. As shown in Table 4, we find that fine-tuning SDXL with data generated with LLaMA3 achieves 78.6% on average on DSG, improving the baseline by 5.3%, closing the gap to GPT-3.5 based results. This demonstrates that SELMA is flexible and compatible with different prompt generator LLMs. Besides, we further experiment with fine-tuning SDXL with data generated with both a weaker image generator SDv2 and a weaker prompt generator LLaMA3. Our results show that this model achieves similar performance as the model fine-tuned with images generated with SDXL, demonstrating that weak-to-strong generalization holds with weaker data generators.

## 5.6 Human Evaluation

In addition to automatic evaluation using text faithfulness benchmarks (DSG and TIFA) and human preference metrics (PickScore, ImageReward, and HPS), we further perform a human evaluation to compare the performance of SDXL and SDXL fine-tuned with SELMA on DSG$^{SELMA-5K}$ (details in Sec. 4.3). We randomly select 200 prompts from DSG and ask three annotators to determine "Which image aligns with the caption better?" given the text prompt and generated images from both SDXL and SDXL+SELMA. We provide win/tie/lose options to the annotators, and we report the win *vs.* lose percentage in the following. The user interface, instructions, and the detailed statistics are provided in appendix.

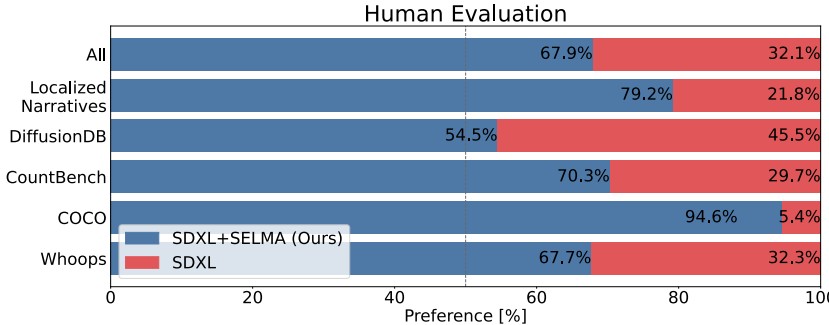

Figure 4: Human Evaluation on 200 sampled text prompts from DSG, where we show the win *vs.* lose percentages of SDXL and SDXL+SELMA (Ours).

**SDXL+SELMA is preferred than SDXL in terms of text alignment.** Fig. 4 shows that on all five DSG splits, images generated with SDXL+SELMA are preferred **67.9%** of the time, compared to 32.1% for the baseline SDXL. Furthermore, on the five datasets fine-tuned with similar text prompts, SDXL+SELMA achieve a preference rate of **94.6%** on COCO split and **79.2%** on Localized Narratives. This substantial preference over the baseline model demonstrates the effectiveness of SELMA in enhancing T2I models.

Table 5: Comparison with different fine-tuning methods on SD v2 with our auto-generated data, in text faithfulness and human preference. See Sec. 5.7 for discussion.

| No. | Methods | Text Faithfulness | | Human Preference on DSG | | |
|---|---|---|---|---|---|---|
| | | DSG$^{\text{mPLUG}}$ ↑ | TIFA$^{\text{BLIP2}}$ ↑ | PickScore ↑ | ImageReward ↑ | HPS ↑ |
| 0. | SDv2 | 70.3 | 79.2 | 20.8 | 0.17 | 24.0 |
| 1. | + LoRA Merging (SELMA) | **77.7** | **83.2** | **21.3** | **0.72** | **27.5** |
| 2. | + LoRA Merging + DPO | 75.1 | 81.4 | 20.8 | 0.44 | 26.0 |
| 3. | + MoE-LoRA | 77.2 | 83.0 | **21.3** | 0.68 | 27.2 |

## 5.7 Training Method Ablations

We experiment with various training configurations for SELMA to validate our design choices for fine-tuning. As our current experiments are based on supervised fine-tuning with LoRA Merging, we additionally explore Direct Preference Optimization (DPO) [55; 76] as an alternative to supervised fine-tuning and employing Mixture of Lora Experts (MoE-LoRA) [80] instead of LoRA Merging. See the Appendix for the implementation details. Table 5 demonstrates that while fine-tuning with DPO and MoE-LoRA significantly improves the T2I models' text faithfulness and human preference (*No.2 & 3. vs. No.0.*), simple inference-time LoRA merging achieves the best overall performance. In the end, we adopt LoRA merging and supervised fine-tuning as the default configuration in SELMA for its simplicity and efficiency.

## 5.8 LoRA Expert Size Ablations

In this section, we demonstrate that simply scaling the LoRA size doesn't help mitigate the knowledge conflict between different skills. We experiment with using a single LoRA with different ranks (128 / 256 / 640) and compare them to our default LoRA merging (rank=128). Table 6 shows that increasing the rank of LoRA from 128 to 256 slightly improves the performance (i.e., 74.9 vs. 74.4), but further scaling the rank of the LoRA to 640 significantly drops

Table 6: Performance of scaling the LoRA ranks on DSG.

| Model | LoRA Rank | DSG$^{\text{mPLUG}}$ ↑ |
|---|---|---|
| Base model (SDv2) | - | 70.3 |
| Single LoRA | 128 | 74.4 |
| Single LoRA | 256 | 74.9 |
| Single LoRA | 640 | 71.5 |
| LoRA Merging | 128 | 77.7 |

the performance (i.e., 71.5 vs. 74.4). The performance drop when using LoRA with higher ranks (i.e., rank=640) is similar to the observation in Figure 3 in [24]. This result indicates the effectiveness of our skill-specific learning and merging of LoRA experts.

| SDXL | SDXL+SELMA | SDXL | SDXL+SELMA |

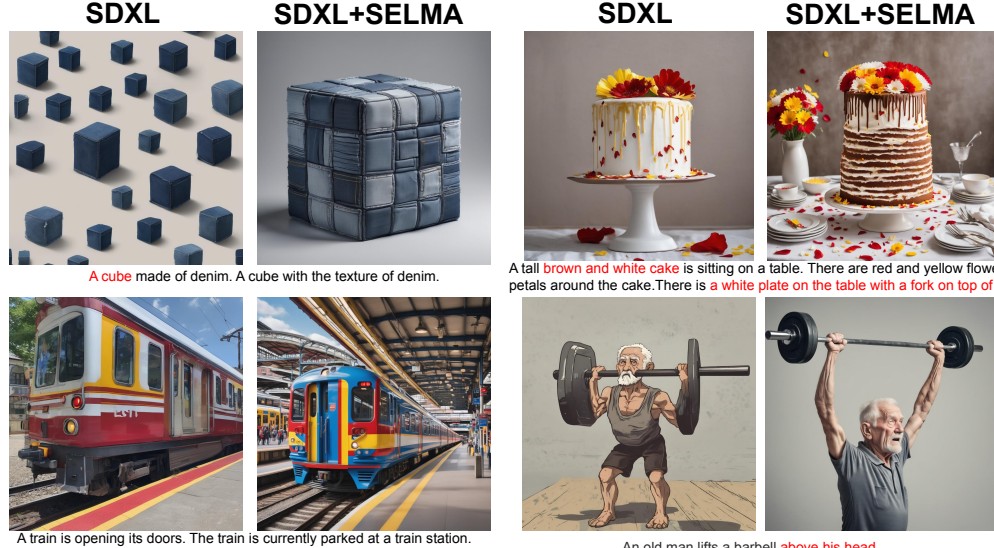

Figure 5: Example images generated with SDXL and SDXL+SELMA. SELMA shows better performance in object composition, attribute binding, and long text prompt following. We highlight the parts of the prompts in red where SDXL makes errors while SDXL+SELMA generates correctly.

## 5.9 Qualitative Examples

We show some qualitative examples of images generated with SDXL fine-tuned with SELMA paradigm in Fig. 5. We find that fine-tuning with SELMA improves SDXL's capability in composing infrequently co-occurred attributes (*i.e.*, "cube" and "denim" in the top-left image), composing multiple objects mentioned in the text prompts (*i.e.*, "brown and white cake", "table", "red and yellow flower", and "fork" in the top-right image), following details in the long paragraph-style text prompts (*i.e.*, "blue train with white stripes" and "long yellow line near train area" in the bottom-left image), and generating images that challenge commonsense (*i.e.*, "Old man lifts a barbell" in bottom-right image). These qualitative examples demonstrate the effectiveness of SELMA in improving T2I models' text faithfulness and human preference.

## 6 Conclusion

We propose SELMA, an novel paradigm to improve state-of-the-art T2I models' faithfulness in generation and human preference by eliciting the pre-trained knowledge of T2I models. SELMA first collects self-generated images given diverse generated text prompts without the need for additional human annotation. Then, SELMA fine-tunes separate LoRA models on different datasets and merges them during inference to mitigate knowledge conflict between datasets. SELMA demonstrates strong empirical results in improving T2I models' faithfulness and alignments to human preference and suggests potential weak-to-strong generalization for diffusion-based T2I models.

## Acknowledgement

We thank the reviewers and Elias Stengel-Eskin, Prateek Yadav, and Yushi Hu for the thoughtful discussion. This work was supported by DARPA ECOLE Program No. HR00112390060, NSF-AI Engage Institute DRL-2112635, DARPA Machine Commonsense (MCS) Grant N66001-19-2-4031, ARO Award W911NF2110220, ONR Grant N00014-23-1-2356, and a Bloomberg Data Science Ph.D. Fellowship. The views contained in this article are those of the authors and not of the funding agency.

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

# Appendix

In this appendix, we present the following:

- Details of evaluation metrics we use (Appendix A).

- Evaluation on HPS v2.1 benchmark (Appendix B).

- Additional qualitative examples with SELMA on SDXL backbone (Appendix C).

- Skill-specific VQA accuracy on both TIFA and DSG benchmarks (Appendix D).

- Human evaluation details (Appendix E).

- Additional comparison with fine-tuning with filtered generated data (Appendix F).

- Bias evaluation in fine-tuned T2I models (Appendix G).

- Implementation details of SELMA (Appendix H).

- Implementation details of two training configuration variants: DPO and MoE-LoRA (Appendix I).

- Prompts we used to query LLM to generate new text data (Appendix J).

- Limitations and broader impact of SELMA approach (Appendix K).

- License information for data and model used in this paper (Appendix L).

## A  Evaluation Metrics

We quantitatively evaluate the performance of T2I generation models in text faithfulness and human preference metrics.

**Text faithfulness.** To evaluate T2I model's faithfulness in generation, we use VQA accuracy from TIFA and DSG. Specifically, TIFA and DSG utilize LLMs to generate questions given a text prompt and utilize the VQA model to check whether it can answer the questions correctly given the generated image. The image is considered to have better faithfulness to text prompts if the VQA model can answer the question more correctly. For TIFA, we use BLIP-2 as the VQA model following Sun *et al*. [70], For DSG, we use mPLUG-large [35] as the VQA model, as PaLI [12] is not publicly accecssible, and Hu *et al*. [28] shows that mPLUG achieves higher human correlation than BLIP-2.

**Human preference metrics.** To evaluate how the generated images align with human preference, we use the PickScore [31], ImageReward [82], and HPS [79]. PickScore and HPS are based on CLIP [54] trained on the Pick-a-Pic dataset [31] and Human Preference Score dataset [79] respectively, which both have annotations of human preference over images. ImageReward is a BLIP [36] based reward model fine-tuned on human preference data collected on DiffusionDB. We calculate PickScore, ImageReward, and HPS on the 1060 DSG prompts. We also provide the evaluation results on HPS prompts in the appendix.

## B  Evaluation on HPS v2.1 Benchmark

In the main paper, we calculate HPS score [79] on text prompts on DSG benchmark, following DreamSync [70]. In this section, we additionally show the HPS score on the prompts from HPS v2.1 benchmark. HPS benchmark contains 3200 unique prompts from four different categories: anime, concept-art, paintings, and photo. We calculate the HPS score based on its HPS v2.1 model trained on higher quality datasets. As shown in Table 7, when adapting SELMA to different stable diffusion base model, our approach significantly improves the baseline performance (*i.e*., 2.5 for SD v1.4, 4.3 for SD v2, and 1.4 for SDXL), achieving better performance than all the released model on the HPS benchmark.[2]

---

[2]Benchmark performance can be found: `https://github.com/tgxs002/HPSv2`

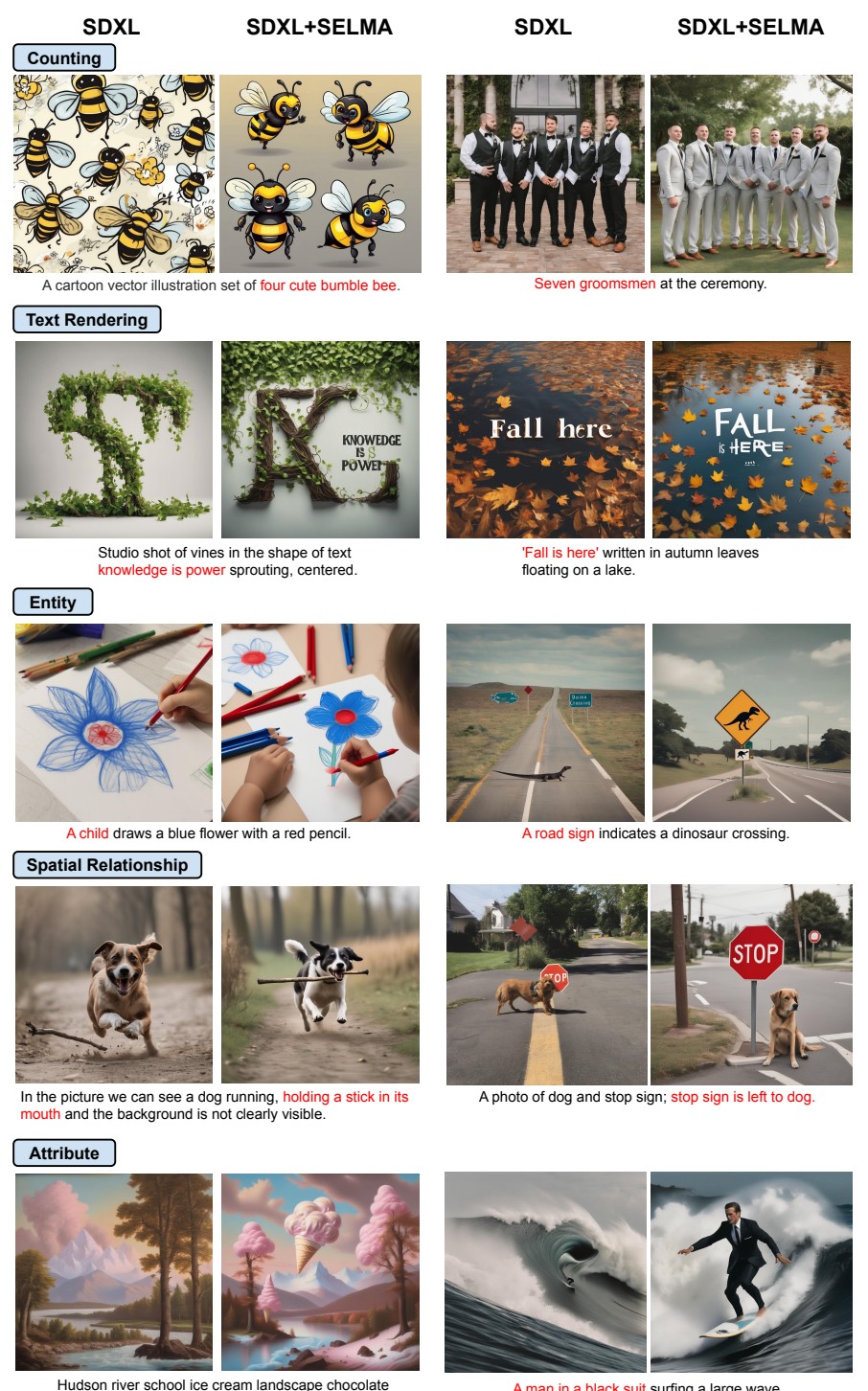

Figure 6: Qualitative example images generated with SDXL and SDXL+SELMA (Ours) from DSG [14] test prompts requiring different skills. SELMA helps improve SDXL in various skills, including counting, text rendering, spatial relationships, and attribute binding. We highlight the parts of the prompts in red where SDXL makes errors while SDXL+SELMA generates correctly.

Table 7: Evaluation on HPS v2.1 evaluation benchmark. SELMA achieves signifiantly better scores on HPS evaluation benchmark compared with baselines, and outperforming all other baselines reported in HPS v2.1 benchmark.

| Method | HPS v2.1 Evaluation Benchmark | | | | |
|---|---|---|---|---|---|
| | Anime | Concept-Art | Paintings | Photo | Average |
| SD v1.4 [59] | 26.0 | 24.9 | 24.8 | 25.7 | 25.4 |
| + SELMA | **28.2** | **27.6** | **27.8** | **28.0** | **27.9** |
| SD v2 [59] | 27.1 | 26.0 | 25.7 | 26.7 | 26.4 |
| + SELMA | **32.0** | **30.3** | **30.0** | **30.4** | **30.7** |
| SDXL [50] | 33.3 | 32.1 | 31.6 | 28.4 | 31.3 |
| + SELMA | **34.7** | **32.7** | **32.6** | **30.8** | **32.7** |

Table 8: Detailed skill-specific comparison of SD models *vs*. SD models+SELMA on TIFA benchmark.

| Method | TIFA skills | | | | | | | | | | | | |
|---|---|---|---|---|---|---|---|---|---|---|---|---|---|
| | Animal/Human | Object | Location | Activity | Color | Spatial | Attribute | Food | Counting | Material | Other | Shape | Average |
| SD v1.4 [59] | 83.7 | 78.3 | 80.3 | 71.7 | 73.0 | 58.9 | 74.5 | 81.8 | 63.3 | 76.6 | 47.3 | **65.2** | 75.8 |
| + SELMA | **87.1** | **83.0** | **84.8** | **75.9** | **74.4** | **62.3** | **76.0** | **88.1** | **66.2** | **78.5** | **52.2** | 59.4 | **79.5** |
| SD v2 [59] | 86.5 | 82.6 | 83.8 | 75.6 | 76.8 | 62.4 | 75.4 | 85.2 | 66.5 | **82.2** | 55.4 | **75.0** | 79.2 |
| + SELMA | **89.7** | **88.0** | **87.6** | **80.3** | **80.3** | **66.0** | **77.2** | **91.0** | 65.8 | 81.3 | **63.2** | 68.1 | **83.2** |
| SDXL [50] | 90.3 | 86.4 | 86.6 | 80.0 | 78.6 | 67.7 | 78.3 | 90.6 | 67.4 | **84.2** | 67.7 | **62.3** | 82.9 |
| + SELMA | **93.4** | **90.4** | **89.5** | **83.6** | **81.1** | **69.6** | **78.5** | **92.1** | **68.8** | 83.7 | **68.7** | 60.9 | **85.6** |

# C  Additional Qualitative Exmaples

In Fig. 6, we show additional qualitative examples of SDXL and SDXL+SELMA from DSG [14] test prompts requiring different skills. SELMA helps improve SDXL in various skills, including counting, text rendering, spatial relationships, and attribute binding. For counting skill prompts, SDXL+SELMA generates "four bees" and "seven groomsmen" correctly following the text prompts. For text rendering skill prompts, SDXL+SELMA can render the text ("knowledge is power" and "Fall is here") more accurately, while it still lacks the capability to render the text in the texture of vines or autumn leaves. For entity skill (placing correct objects) prompts, the SDXL sometimes misses some entities mentioned in the text prompt (*i.e*., "A child", and "A road sign"), while SDXL+SELMA can successfully generate them. For spatial relationship skill prompts, SDXL+SELMA generated images (*i.e*., "holding a stick in its mouth", and "stop sign left to dog"). Lastly, for attribute skill prompts, SDXL+SELMA binds objects with their corresponding attributes (*i.e*., "cotton candy trees" and "A man in black suit") more accurately than SDXL. These qualitative results demonstrate the effectiveness of SELMA.

# D  Skill-specific VQA Accuracy on TIFA and DSG

In this section, we show the detailed VQA accuracy for each skill category on TIFA and DSG benchmarks. Since DreamSync [70] does not provide the skill-specific scores, we report the skill-specific scores of SD models on TIFA and DSG and based on our experiments in Table 8 and Table 9; we observe there are less than 1% score differences of SD/SDXL models in TIFA average accuracy between the results in Sun *et al*. and ours.

As shown in Table 8 and Table 9, on both TIFA and DSG benchmarks, SELMA improves the generation faithfulness in most of the categories. Comparing SDXL and SDXL+SELMA, the SDXL finetuned with SELMA approach shows large improvement especially in entity (*i.e*., 3.1% on animal/human on TIFA compared with SDXL, 5.6% in whole on DSG, 12.2% in part on DSG), as well as spatial relationship (*i.e*., 1.9% on TIFA, and 8.0% on DSG), and counting skills (*i.e*., 1.4% on TIFA, and 13.2% on DSG). Besides, we also observe that SELMA significantly improves the text rendering for SD v2 and SDXL (*i.e*., 16.4% compared with SDXL, and 6.5% compared with SD v2 on DSG), but not SD v1.4 (*i.e*., 1.8% decrease compared with SD v1.4 on DSG).

Table 9: Detailed skill-specific comparison of SD models *vs.* SD models+SELMA on DSG benchmark. We show the skill categories that have more than 50 questions.

| Method | DSG skills | | | | | | | | | | | | |
|---|---|---|---|---|---|---|---|---|---|---|---|---|---|
| | Whole | Color | Shape | Spatial | Part | State | Count | Action | Global | Material | Type | Text Rendering | Average |
| SD v1.4 [59] | 78.6 | **62.5** | 46.0 | 61.1 | 68.1 | 58.2 | 62.4 | 59.9 | **59.4** | 42.3 | **73.0** | **52.7** | 67.2 |
| + SELMA | **83.7** | **62.5** | **52.0** | **66.2** | **72.1** | **63.3** | **66.1** | **72.1** | 57.1 | **59.7** | 67.6 | 50.9 | **71.3** |
| SD v2 [59] | 80.8 | 68.6 | 50.0 | 63.6 | 72.3 | 63.6 | **69.3** | 62.9 | **61.9** | 55.7 | 66.8 | 60.9 | 70.3 |
| + SELMA | **88.0** | **80.8** | **65.4** | **71.0** | **78.7** | **71.5** | 66.3 | **78.4** | 61.0 | **69.2** | **81.4** | **67.4** | **77.7** |
| SDXL [50] | 84.8 | 74.7 | 58.0 | 69.4 | 71.1 | 60.7 | 59.8 | 71.7 | **61.5** | 63.9 | 71.7 | 60.0 | 73.3 |
| + SELMA | **90.4** | **81.3** | **64.0** | **77.4** | **83.3** | **68.2** | **73.0** | **79.4** | 60.2 | **77.3** | **75.4** | **76.4** | **80.2** |

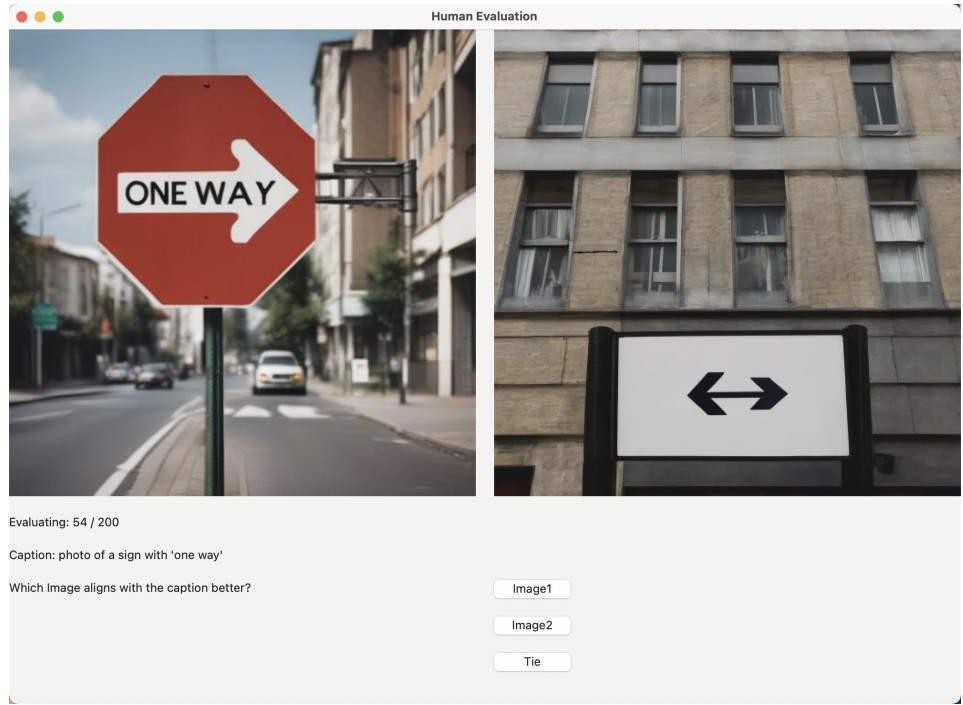

Figure 7: Example user interface for human evaluation on DSG prompts.

# E    Human Evaluation Details

We conduct the human evaluation (described in the main paper Sec. 5.5) on 200 randomly sampled prompts from DSG, with three external annotators. We show the annotation interface in Fig. 7. The image order between the two models is randomized to avoid the leakage of information about which image is generated with which model.

In Table 10, we show the detailed annotator votes for win, lose, and tie for SDXL and SDXL+SELMA. SDXL+SELMA has significantly higher win votes compared with SDXL on all the 200 sampled text prompts (*i.e.*, 241 win *vs.* 114 lose), demonstrating the effectiveness of SELMA.

Table 10: Human Evaluation on 200 sampled text prompts from DSG. We show the detailed win/lose/tie counts on all samples and samples from each dataset.

| Eval Dataset | Win | Lose | Tie |
|---|---|---|---|
| All | 241 | 114 | 245 |
| Localized Narratives | 19 | 5 | 12 |
| DiffusionDB | 6 | 5 | 34 |
| CountBench | 26 | 11 | 17 |
| COCO | 35 | 2 | 47 |
| Whoops | 21 | 10 | 26 |

# F    Comparison with Fine-tuning with Filtered Generated Data

We demonstrate that our automatically collected data is of high quality to improve the faithfulness of T2I models. Specifically, we filter out the generated images based on TIFA score ($>0.9$) and VILA score ($>0.6$), and maintain 1K data for each set of skill-specific prompts. Fine-tuning with

filtered images achieves 80.3% DSG score with SDXL backbone, which is similar performance as fine-tuning with un-filtered version (i.e., 80.2% on DSG), indicating the overall high quality of the data, which does not lead to performance degradation.

## G   Bias Evaluation in T2I Models

In this section, we investigate whether fine-tuning with T2I-generated images will increase the bias in existing T2I models. Specifically, we evaluate the T2I model with GEP score [90], which measures the gender bias in T2I models. Specifically, a learned cross-modal classifier is used to distinguish attributes (*e.g.*, dress) in the image, and GEP score calculates attribute-wise difference based on output score from the cross-modal classifier. A lower GEP score indicates less gender bias. Fine-tuning SD v2 with our generated data achieves 0.04413 GEP score, while fine-tuning with ground truth data achieves 0.04976. This indicates that fine-tuning with our generated data does not amplify the bias in existing T2I models compared with fine-tuning with ground truth data.

## H   Implementation Details of SELMA

In the **prompt generation** stage (Sec. 3.1), we use `gpt-3.5-turbo-instruct` [46] to generate text prompts by providing three prompts for each skill as in-context examples. For each of the five datasets (COCO [39], Localized Narratives [51], DiffusionDB [77], CountBench [47], and Whoops [6]), we collect 1K prompts starting with three prompts randomly sampled from them, ensuring the prompts are not included in the DSG test prompts (*i.e.*, 5K prompts in total). We refer to the resulting auto-generated datasets as Localized Narrative$^{\text{SELMA}}$, CountBench$^{\text{SELMA}}$, DiffusionDB$^{\text{SELMA}}$, Whoops$^{\text{SELMA}}$, and COCO$^{\text{SELMA}}$. We refer to the resulting combination of 5K auto-generated dataset as DSG$^{\text{SELMA-5K}}$.

In the **image generation** stage (Sec. 3.2), we use the default denoising steps 50 for all models, and the Classifier-Free Guidance (CFG) [26] of 7.5. In the **LoRA fine-tuning** stage (Sec. 3.3), we use 128 as the LoRA rank. We fine-tune LoRA in mixed precision (*i.e.*, FP16) with a constant learning rate of 3e-4 and a batch size of 64. We fine-tune LoRA modules for 5000 steps, which is approximately 313 epochs. **During inference**, we uniformly merge the specialized LoRA experts into one multi-skill expert (Sec. 3.4). We evaluate model checkpoints every 1000 steps and pick the model with the best text faithfulness on DSG benchmark. Fine-tuning LoRA for SD v1.4, SD v2, and SDXL takes 6 hours, 6 hours, and 12 hours on a single NVIDIA L40 GPU, respectively. We use Diffusers [49] for our experiments.

## I   Implementation Details of DPO and MoE-LoRA

In this section, we provide the implementation details of two training approaches we experiment with (described in Sec. 5.6 in the main paper).

**Direct Preference Optimization (DPO).** We fine-tune LoRA models with DPO proposed in [76]. Specifically, we sample two images with T2I models and calculate the image-text alignment with CLIP score [54]. We use the image with a higher CLIP score as the positive example and the image with a lower CLIP score as the negative example. We fine-tune the LoRA models to learn to generate images closer to the positive image distribution and push away from generating images similar to the negative image distribution. Similarly, we fine-tune with DPO on five datasets with different text styles and skills and merge LoRA expert models during inference time by averaging the LoRA weights. In DPO training, we use a constant learning rate 3e-4 and fine-tune LoRA for 5K steps. We evaluate the model on DSG every 1K steps and pick the best checkpoint.

**Mixture of Lora Experts (MoE-LoRA).** MoE-LoRA [80] utilizes a gating function (router) to decide which experts to use during training and inference. The gating function predicts weights for each expert based on layer inputs and picks the top $K$ experts to use at each layer. Specifically, the gating function we use is a simple linear mapping function, where $\{w_i\}_{i=1}^{K} = W_g x$. $x$ is the input to each layer, $W_g$ is the learnable gating weights, and $\{w_i\}_{i=1}^{K}$ are the predicted weights. The outputs of each expert are added together with the normalized weights from the gating function. In MoE-LoRA, we initialize five LoRA experts fine-tuned on different datasets containing different text styles and skills. We freeze the learned LoRA weights and only fine-tune the gating function

on the collected five datasets. We also compare with learning LoRA weights along with the router, which achieves worse performance (75.9 on DSG). We activate all five experts during training and inference (*i.e.*, $K = 5$). In MoE-LoRA training, we use a constant learning rate 1e-5 and fine-tune LoRA for 5K steps. We evaluate the model on DSG every 1K steps and pick the model with highest text faithfulness score.

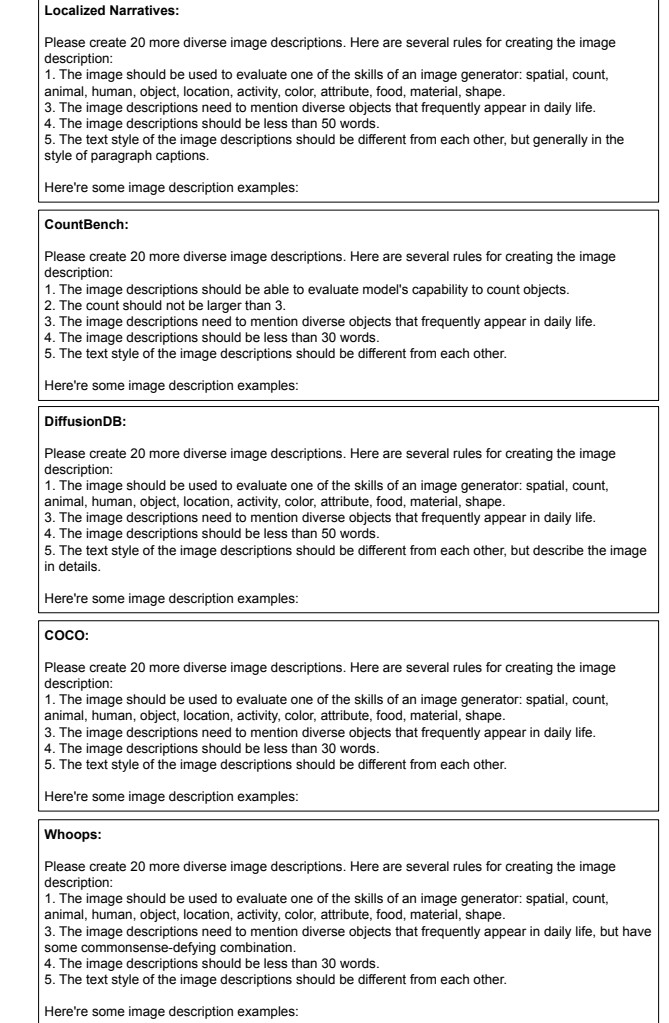

Figure 8: Prompts used to query GPT-3.5 auto-generate new prompts targeting different skills.

## J Skill-Specific Prompt Generation Details

We show the prompts we use to query GPT3.5 to generate 1K prompts for each skill. As shown in Fig. 8, we use different prompts to generate SELMA data. For example, we specify "paragraph captions" to generate text prompts that can be used to teach model to follow long text prompts, and specifying "evaluate model's capability to count objects" to collect a set of prompts for improving model's counting capability. Besides, in all the prompt generation, we emphasize that the image should "mention diverse objects" to maximize the semantic diversity in generated prompts.

# K  Limitations and Broader Impact

Text-to-image generation models can be used in many real-world applications, such as creating content for media and entertainment. Our proposed SELMA improves T2I models' faithfulness with auto-generated data, reducing human annotation efforts for high-quality image-text pairs.

However, we also note that our SELMA has several limitations. SELMA relies on a strong image generator and an instruction-following LLM. Note that SELMA is model-agnostic and can be implemented with publicly accessible models (GPT-3.5/LLaMA3 and Stable Diffusion models). Also, since our fine-tuning works well with a small number of image-text pairs (*i.e.*, for each skill, we only generate 1K text prompts and generating one image per each text prompt), the cost of LLM inference (*i.e.*, $27.78 for querying GPT-3.5 for generating prompts in all the experiments including ablation studies) and image generation (8s per image for image generation with SDXL on a single NVIDIA L40 GPU with 48GB memory) is minimal. Besides, although SELMA helps boost T2I models' performance significantly in both text faithfulness and alignment to human preference, fine-tuning with SELMA does not guarantee the resulting model to follow the text prompts in every detail. To use T2I models trained with SELMA, researchers should first carefully study their capabilities in relation to the specific context in which they are being applied.

# L  Licenses

We provide the licenses of the existing assets we use in this paper in Table 11.

Table 11: A list of the licenses of the existing assets used in this paper.

| Asset | License |
|---|---|
| PyTorch [3] | BSD-style |
| Huggingface Transformers [78] | Apache License 2.0 |
| Torchvision [44] | BSD 3-Clause "New" or "Revised" License |
| Diffusers [49] | Apache License 2.0 |
| Stable Diffusion [59] | CreativeML Open RAIL-M |
| COCO dataset [39] | CC BY 4.0 |
| Localized Narrative dataset [51] | CC BY 4.0 |
| DiffusionDB [77] | MIT License |
| Whoops [6] | CC BY 4.0 |
| CountBench [47] (LAION-400M [64] subset) | CC BY 4.0 |
| GPT3.5 [46] | OpenAI Terms of Use |
| LLaMA3 [1] | Meta LLaMA3 License |

