# OpenReview forum: "SELMA: Learning and Merging Skill-Specific Text-to-Image Experts with Auto-Generated Data"
_NeurIPS.cc/2024/Conference — NeurIPS 2024 poster_

### Official Review · Reviewer_WQ94 · 2024-06-28

**Soundness:** 3
**Presentation:** 3
**Contribution:** 2
**Rating:** 6
**Confidence:** 4

**Summary:**

The paper proposes a workflow in which synthetic Text To Image data is generated in order to improve faithfulness of T2I models. Specifically, they generate prompts with LLMs, then Images with T2I models and finally fine-tune pre-trained T2I models with LORA fine-tuning. They fine-tune multiple LORA experts and then merge them by merging parameters. On a limited set of evals, the proposed model achieves good results.

**Strengths:**

The paper addresses the faithfulness of Text To Image models. With the increasing application of T2I models across many domains, faithfulness becomes and increasingly important research topic in both industry as well as the academic community.

I appreciate the discussions of related works and how the paper at hand differs from prior art.

While the paper motivates its contributions with faithfulness, the proposed workflow is actually independent from faithfulness itself. Faithfulness is only a side effect of the proposed method which has the main focus on multi-task fine-tuning for T2I models. As such, the impact could be even bigger, if the method gets applied to other challenges in the field of T2I.

**Weaknesses:**

The strength mentioned above is also a weakness of the paper. The paper is motivated by faithfulness. However, the proposed method tackles faithfulness at best as a side effect. While this means that the method might be more general, it also negatively impacts the clarity of the paper and ease of understanding. It might be better to just focus on the multi-task fine-tuning task instead of the impact of individual tasks or side-effects.

**Questions:**

Given the complex setup of the proposed workflow, there is a large number of design decisions to make. Do you have guidance, experiments which part of the workflow is the most sensitive with respect to downstream results, e.g., LLM choice, in-context example selection, LLM prompt choice, T2I choice, LORA parameters, etc.?

**Limitations:**

The limitations are only discussed in the appendix. I think it would be important to have a comprehensive discussion of limitations in the main text.

---

> ### Author Rebuttal · Authors · 2024-08-06
>
> Thanks for your valuable feedback. Below we address your questions with further clarifications and experiments.
>
> > **W1. The strength mentioned above is also a weakness of the paper. The paper is motivated by faithfulness. However, the proposed method tackles faithfulness at best as a side effect. While this means that the method might be more general, it also negatively impacts the clarity of the paper and ease of understanding. It might be better to just focus on the multi-task fine-tuning task instead of the impact of individual tasks or side-effects.**
>
> We would like to clarify that our framework is not only motivated by text faithfulness but also designed to directly tackle and improve multiple aspects (i.e., skills) of faithfulness. To improve multiple different aspects of faithfulness, our framework (1) automatically generates datasets for teaching faithfulness in different aspects and (2) obtains a T2I model with better text faithfulness, via skill-specific expert learning and expert merging.
>
> > **Q1. Given the complex setup of the proposed workflow, there is a large number of design decisions to make. Do you have guidance, experiments which part of the workflow is the most sensitive with respect to downstream results, e.g., LLM choice, in-context example selection, LLM prompt choice, T2I choice, LORA parameters, etc.?**
>
> Thanks for the question. Below we provide ablation results of two design choices: Skill-specific LoRA merging and prompt generator LLM choices. Some of the analysis are from paper Table 2 and Table 9.
>
> (1) First, we show that skill-specific LoRA merging is crucial for mitigating knowledge conflict and improving text faithfulness of T2I models. Merging skill-specific experts during inference outperforms single-LoRA by 3.3% on the DSG benchmark. Besides, merging skill-specific experts also works better than fine-tuning a single LoRA with more parameters (i.e., larger rank), demonstrating the effectiveness of learning separate LoRA to avoid knowledge conflict.
>
> | Approach     | LoRA Rank | DSG  |
> |--------------|-----------|------|
> | SDv2     |           | 70.3 |
> | Single LoRA  | 128       | 74.4 |
> | Single LoRA  | 256       | 74.9 |
> | Single LoRA  | 640       | 71.5 |
> | LoRA Merging (ours) | 128       | **77.7** |
>
> (2) Second, we show the effectiveness of stronger prompt generator LLMs. We collect 5K skill-specific prompts with Llama-3 and GPT-3.5, and fine-tune SDXL with self-generated images. The table below shows that learning from prompts generated with GPT-3.5 outperforms learning from prompts generated with Llama-3 (80.2 vs. 78.6 on DSG benchmark), demonstrating a stronger LLM can help improve the performance. However, note that both SDXL models fine-tuned with GPT-3.5 and Llama-3 prompts outperform the original SDXL baseline by a large margin, demonstrating the effectiveness of our pipeline even with weaker LLMs.
>
> | Approach | Prompt Generator | DSG  |
> |----------|------------------|------|
> | SDXL | -                | 73.3 |
> | SDXL     | Llama-3          | 78.6 |
> | SDXL     | GPT-3.5          | **80.2** |

---

> > ### Comment · Reviewer_WQ94 · 2024-08-12
> > **Final review**
> >
> > I would like to thank the authors for their response to both my questions and the questions by the fellow reviewers. After going over the other reviews and considering all the answers, I still believe that the contributions and novelty of the paper are sufficient to slightly pass the bar for acceptance.

---

### Official Review · Reviewer_c1fe · 2024-07-09

**Soundness:** 3
**Presentation:** 3
**Contribution:** 3
**Rating:** 6
**Confidence:** 4

**Summary:**

This article goes through four stages: (1) collecting skill-specific prompts using in-context learning of LLMs, (2) self-generating image-text samples for diverse skills without the need for human annotation or feedback from reward models, (3) fine-tuning the expert T2I models on these datasets separately, and (4) obtaining the final model by merging experts from each dataset for efficient adaptation to different skills and mitigation of knowledge conflict in joint training. It is found that the model can optimize itself to an excellent level relying solely on the prompts from the LLM and the model's own generative capabilities.

**Strengths:**

1. This article demonstrates through experiments an interesting conclusion: the model, relying solely on the images generated by prompts, can still be trained on certain domains and achieve superior results without any additional annotation information.

2. This article proves that T2I models struggle with LoRA to accommodate distinct skills and writing styles from different datasets and proposes that using a training-free multi-LoRA fusion method for LoRA trained on different tasks can effectively alleviate this issue.

3. This article demonstrates that even when using data generated by a weaker T2I model, it is still possible to enhance the performance of a stronger model.

**Weaknesses:**

1. The article does not provide a detailed explanation of the comparison in Table 2 regarding single LoRA. In the comparison between multi-LoRA and single LoRA, are the parameter counts of multi-LoRA and single LoRA the same, or is each LoRA within multi-LoRA equivalent in parameter count to single LoRA? Additionally, is the total training step count for multiple LoRAs in multi-LoRA equal to the training step count for single LoRA, or is the training step count for single LoRA consistent with that of each LoRA within multi-LoRA?

2. The fusion mechanism among multiple LoRAs does not seem to have been thoroughly explored through sufficient ablation experiments. If, after training each LoRA separately in multi-LoRA, a router is introduced to perform gating operations on multi-LoRA, similar to MoE-LoRA, would the effectiveness improve?

**Questions:**

See weaknesses.

**Limitations:**

Yes.

---

> ### Author Rebuttal · Authors · 2024-08-06
>
> Thanks for your valuable feedback. Below we address your questions with further clarifications and experiments.
>
> > **W1-1. In the comparison between multi-LoRA and single LoRA, are the parameter counts of multi-LoRA and single LoRA the same, or is each LoRA within multi-LoRA equivalent in parameter count to single LoRA?**
>
> The single LoRA that learns multiple skills has the same number of parameters for each skill-specific LoRA. Furthermore, we experiment with LoRA with more parameters by using higher ranks (256 / 640) and compare with our default LoRA merging (rank=128).
>
>
> | Approach     | LoRA Rank | DSG  |
> |--------------|-----------|------|
> | SDv2    |           | 70.3 |
> | Single LoRA  | 128       | 74.4 |
> | Single LoRA  | 256       | 74.9 |
> | Single LoRA  | 640       | 71.5 |
> | LoRA Merging (ours) | 128       | **77.7** |
>
> The table shows that increasing the rank of LoRA from 128 to 256 slightly improves the performance (i.e., 74.9 vs. 74.4), but further scaling the rank of the LoRA to 640 significantly drops the performance (i.e., 71.5 vs. 74.4). The performance drop when using LoRA with higher ranks (i.e., rank=640) is similar to the observation in Figure 3 in [A]. This result indicates the effectiveness of our skill-specific learning and merging of LoRA experts. We will add this additional result in the final version.
>
> > **W1-2. Additionally, is the total training step count for multiple LoRAs in multi-LoRA equal to the training step count for single LoRA, or is the training step count for single LoRA consistent with that of each LoRA within multi-LoRA?**
>
> We let the T2I model see each example for the same amount of times (i.e., same epochs), in both settings.
>
> For (1) “single LoRA for multiple skill-specific image-text pairs”, we train the LoRA for 25K training steps on 5K image-text pairs with batch size 64.
>
> For (2) “learning skill-specific LoRAs for each skill-specific image-text pair and merging them” (ours), we train each LoRA for 5K training steps on each of 1K image-text pairs with batch size 64.
>
>
> > **W2. If, after training each LoRA separately in multi-LoRA, a router is introduced to perform gating operations on multi-LoRA, similar to MoE-LoRA, would the effectiveness improve?**
>
> As described in L741-744, we freeze the learned LoRA weights and only fine-tune the gating function (i.e., router), which is the setting you mentioned. Additionally, we experiment with learning LoRA weights along with the router. As shown in the below table, learning LoRA weights along with the router achieves worse performance than when LoRAs were frozen. Our default expert merging method – LoRA merging – performs the best. We will add this additional result in the final version.
>
> | Approach         | DSG  |
> |------------------|------|
> | SDv2         | 70.3 |
> | MoE-LoRA (learning router and LoRAs from scratch) | 75.9 |
> | MoE-LoRA (learning router only; LoRAs are frozen)  | 77.2 |
> | LoRA Merging (default)  | **77.7** |
>
> [A] He et al. (2024), "Sparse Matrix in Large Language Model Fine-tuning"

---

> > ### Comment · Reviewer_c1fe · 2024-08-13
> > **Response to authors**
> >
> > The authors solve my issues and I maintain the rating to accept it.

---

### Official Review · Reviewer_xcmQ · 2024-07-09

**Soundness:** 4
**Presentation:** 4
**Contribution:** 3
**Rating:** 7
**Confidence:** 3

**Summary:**

The paper analyzes the merging of skill-specific LoRA-finetuned models, trained on generated data and compares this approach to other training approaches (i.e. finetuning, PPO). Moreover, the paper compares the use of GT images and prompts to the use of generated data. The results suggest that this approach performs better than the baseline approaches. Finally, the paper shows a weak-to-strong generalization from weaker models.

**Strengths:**

The presented approach shows clear advantages over other methods for alignment - both on text faithfulness and human preference. The paper is easy to read and to understand. Both the effectiveness of auto-generated data (text and images) as well as the merging of experts approach is well studied and ablated.

**Weaknesses:**

_Weak-to-Strong Generalization_

The claim in this section doesn't mention the fact that the text here comes from an additional strong model. While the generative model is weaker, the use of an LLM here makes this experiment less convincing. One possible way to improve this section is by ablating here separately again the text and the images.

_Ablation of LoRA parameters_

Will a larger $r$ value for the LoRA remove the need for the merging? The paper can benefit from ablation of the size of the bottleneck in the LoRA, together with an experiment that shows that the merging is better than one larger LoRA, trained on all the generated data together.

_Optimization time comparisons_

An additional concern for the merged LoRAs approach is its optimization time. Can you provide the comparisons for the training time and iteration number for the presented approach and baselines?

**Questions:**

See weaknesses.

**Limitations:**

Limitations are provided.

---

> ### Author Rebuttal · Authors · 2024-08-06
>
> Thanks for your valuable feedback. Below we address your questions with further clarifications and experiments.
>
> > **W1. Weak-to-Strong Generalization**
>
> Regarding experiments with weaker LM, we would like to bring your attention to Table 9, where we experiment with a LLaMA 3 (8B), which is a publicly available LM known as weaker than GPT-3.5. We find that fine-tuning SDXL with data generated with LLaMA 3 achieves 78.6% on average on DSG, improving the baseline by 5.3% on DSG, closing the gap to GPT-3.5 based results. This demonstrates that SELMA is flexible and compatible even with weaker (but publicly available) prompt generator LMs. Besides, we further experiment with fine-tuning SDXL with data generated with both a weaker image generator SDv2 and a weaker prompt generator LLaMA3. Our results in the table below show that this model achieves similar performance as the model fine-tuned with images generated with SDXL, demonstrating that weak-to-strong generalization holds with weaker data generators.
>
>
> | Base Model | Prompt Generator | Image Generator | DSG  |
> |------------|------------------|-----------------|------|
> | SDXL       | -                | -               | 73.3 |
> | SDXL       | Llama3           | SDv2            | 78.0 |
> | SDXL       | Llama3           | SDXL            | 78.6 |
> | SDXL       | GPT3.5           | SDv2            | 81.3 |
> | SDXL       | GPT3.5           | SDXL            | 80.2 |
>
>
>
> > **W2. Ablation of LoRA parameters**
>
> Following your suggestion, we experiment with LoRA with different ranks (128 / 256 / 640) and compare with our default LoRA merging (rank=128).
>
>
> | Approach     | LoRA Rank | DSG  |
> |--------------|-----------|------|
> | SDv2    |           | 70.3 |
> | Single LoRA  | 128       | 74.4 |
> | Single LoRA  | 256       | 74.9 |
> | Single LoRA  | 640       | 71.5 |
> | LoRA Merging (ours) | 128       | **77.7** |
>
>
> The table shows that increasing the rank of LoRA from 128 to 256 slightly improves the performance (i.e., 74.9 vs. 74.4), but further scaling the rank of the LoRA to 640 significantly drops the performance (i.e., 71.5 vs. 74.4). The performance drop when using LoRA with higher ranks (i.e., rank=640) is similar to the observation in Figure 3 in [A]. This result indicates the effectiveness of our skill-specific learning and merging of LoRA experts. We will add this additional result in the final version.
>
> > **W3. Optimization time comparisons**
>
> In short, there is no meaningful time difference between training (1) single LoRA for multiple skill-specific image-text pairs vs. (2) learning skill-specific LoRAs for each skill-specific image-text pair and merging them, since we let the T2I model see each example for the same amount of times (i.e., same epochs), in both settings.
>
> For (1) “single LoRA for multiple skill-specific image-text pairs”, we train the LoRA for 25K training steps on 5K image-text pairs, which takes around 30h on one single L40 GPU.
>
> For (2) “learning skill-specific LoRAs for each skill-specific image-text pair and merging them” (ours), we train each LoRA for 5K training steps on each of 1K image-text pairs. As shown in Appendix L721-722, this takes around 6h x 5 = 30h on one single L40 GPU (if run parallel in 5 processes, this actually takes 1/5 times of (1)). The LoRA merging takes 26s and is used only once when loading the model before inference, which is negligible compared to the training time.
>
> [A] He et al. (2024), "Sparse Matrix in Large Language Model Fine-tuning"

---

> > ### Comment · Reviewer_xcmQ · 2024-08-09
> >
> > Thank you for the clarifications. I decided to keep my score.

---

### Official Review · Reviewer_dhqF · 2024-07-16

**Soundness:** 3
**Presentation:** 3
**Contribution:** 3
**Rating:** 6
**Confidence:** 3

**Summary:**

The goal of this paper is to improve the faithfulness of text-to-image generation models. New datasets and fine-tuning frameworks are introduced to address this limitation. Specifically, this paper first adopts LLMs to generate multiple datasets of text prompts that can teach different skills and then generates images with a T2I image based on the text prompts. In the second stage, the LoRA finetuning is performed on the generated datasets to get different experts. Finally, different LoRA experts are merged into one better T2I model. The experiments show better performance on public benchmark datasets.

**Strengths:**

1. The experimental results show consistent improvement over multiple base models.

2. The qualitative results look convincing. The proposed model seems to be more faithful than the baselines.

**Weaknesses:**

1. The technical novelty is a bit weak. LoRA fine-tuning and merging experts are not original. The prompt generation seems to be straightforward too.

2. Clarity of this paper should be improved. What is the definition of "Skill"? Is it referred to different prompt styles? Why do the self-generated images improve the T2I model, especially when the T2I model fails to generate faithful images?

**Questions:**

See the weakness.

**Limitations:**

Limitations are discussed in the appendix!

---

> ### Author Rebuttal · Authors · 2024-08-06
>
> Thanks for your valuable feedback. Below we address your questions with further clarifications.
>
> > **W1: The technical novelty is a bit weak. LoRA fine-tuning and merging experts are not original. The prompt generation seems to be straightforward too.**
>
> We would like to first clarify that our main contribution is **introduction of a framework that improves T2I models’ text faithfulness with automatically generated skill-specific data and merge of skill-specific experts** (L40-47). Although LoRA fine-tuning / expert merging / prompt generation methods were initially proposed by previous works, they were not used for improving text faithfulness for T2I models in different skills. Moreover, we would like to elaborate our technical contributions (in relation to previous works):
> - **1. We propose a novel method to automatically create skill-specific image-text pairs based on LLM and the target T2I model itself.** Previous works collect image-text pairs via human annotations (L90-98), and our method significantly reduces the cost for data collection that used to be done via human annotation. Compared to concurrent work relying on heavy image filtering (DreamSync [69]), our data generation method is significantly more efficient by using only 2% of image-text pairs (L98-104).
> - **2. We introduce skill-specific T2I expert learning and merging.** Previous works uses LoRA based T2I model fine-tuning (DreamSync [69]) does not use skill-specific expert learning and merging. ZipLoRA [65] merges two LoRAs for T2I model, but their two LoRAs are limited to a specific subject (e.g., a dog) and a style (e.g., watercolor painting), while we are the first to show the effectiveness of LoRA merging on multiple diverse skills (from 5 datasets) in T2I models (L164-167).
> - **3. We provide comprehensive experiments**, including improvements in two text faithfulness benchmarks (TIFA/DSG), three human preference metrics (Pick-a-Pic/ImageReward/HPS) across three T2I backbones (SD v1.4/v2/XL), human evaluation, ablation studies on design choices, and weak-to-strong generalization (L68-79).
>
> > **W2-1: What is the definition of "Skill"? Is it referred to different prompt styles?**
>
> As we give examples in Fig 1., L121-L122, and L177-178, we use “skills” to refer to different aspects of text prompts that require different writing styles or reasoning capabilities. This includes understanding common objects (e.g., puppy in a backyard), handling long prompts (e.g,. an elegant room with floor-to-ceiling bookshelves, filled with an impressive collection of books of all genres. The cozy reading nook by the window invites anyone to curl up with a good book.”), and displaying commonsense-defying scenes (e.g., cat flying over sky). Following your suggestion, we will add the definition of skill in the introduction section.
>
> > **W2-2: Why do the self-generated images improve the T2I model, especially when the T2I model fails to generate faithful images?**
>
> As we described in L136-142, we conjecture that the T2I model have seen many prompts that require different skills during pre-training, but the T2I model does not have incentive to demonstrate such skills as it is not important in optimizing loss during the pre-training stage. Our fine-tuning stage aims to efficiently extract the knowledge, which we believe was already inside the T2I models, with automatically generated skill-specific image-text pairs.

---

> > ### Comment · Reviewer_dhqF · 2024-08-13
> >
> > The rebuttal addressed my concerns as well as other reviewers' concerns. Therefore, I would like to increase the rating to be a Weak Accept!

---

### Author Rebuttal · Authors · 2024-08-06

We thank the reviewers for their valuable feedback. We also appreciate that they acknowledge SELMA's strengths:
- Clear advantages over other methods for text-image alignment (Reviewer dhqF, xcmQ, WQ94)
- Our proposed automatic data generation pipeline and LoRA merging approach are well studied and ablated (Reviewer xcmQ, c1fe)
- Interesting findings supported with experiments (e.g., weak-to-strong generalization, learning from self-generated images) (Reviewer c1fe).

We have addressed all the questions in our rebuttal, and will incorporate the feedback in the final version.

---

### Decision · Program_Chairs · 2024-09-25

**Decision:**

Accept (poster)

**Comment:**

AC, along with the reviewers, holds a positive opinion towards SELMA. Although the novelty level is being questioned at first, the authors made good attempt to argue this way of using LLM's capability to conduct skill specific text 2 image data augmentation is unique and novel. While presentation could be improved, as suggested by multiple reviewers, AC thinks the current way of presentation, as it is simply a very engineering idea, is already decent. AC strongly suggests the authors take the review comments, especially WQ94's weakness point, into the final draft's revision. Overall, I would recommend Accept (poster).